# PML Bodies in Mitosis

**DOI:** 10.3390/cells8080893

**Published:** 2019-08-14

**Authors:** Anna Lång, Emma Lång, Stig Ove Bøe

**Affiliations:** Oslo University Hospital, Department of Molecular Microbiology, Forskningsveien 1, 0373 Oslo, Norway

**Keywords:** Promyelocytic leukemia protein, PML, PML bodies, APL, mitosis, nuclear import, nuclear bodies, liquid-liquid phase separation

## Abstract

Promyelocytic leukemia (PML) bodies are dynamic intracellular structures that recruit and release a variety of different proteins in response to stress, virus infection, DNA damage and cell cycle progression. While PML bodies primarily are regarded as nuclear compartments, they are forced to travel to the cytoplasm each time a cell divides, due to breakdown of the nuclear membrane at entry into mitosis and subsequent nuclear exclusion of nuclear material at exit from mitosis. Here we review the biochemical and biophysical transitions that occur in PML bodies during mitosis and discuss this in light of post-mitotic nuclear import, cell fate decision and acute promyelocytic leukemia therapy.

## 1. Introduction

The promyelocytic leukemia (PML) gene is located on the long arm of chromosome 15 and was first described as part of a recurrent chromosomal translocation found in more than 95% of patients diagnosed with acute promyelocytic leukemia (APL) [1,2,3]. It contains 9 exons and can be subjected to alternative mRNA splicing, leading to the expression of at least seven main PML isoforms. All PML variants are structurally organized into an N-terminal tripartite (TRIM) domain and a variable C-terminal sequence [4]. The TRIM domain is shared by a superfamily of approximately 60 different human TRIM proteins, several of which have the ability to self-associate into protein networks that occupy discreet cellular compartments [5]. In the case of PML, TRIM is essential for the formation of nuclear structures called PML bodies. 

The PML protein has been implicated in diverse cellular processes ranging from tumor suppression to defense against virus infection [6,7,8]. Although PML-deficient mice originally were described as being grossly indistinguishable from wild type littermates with respect to appearance, growth rate and lifespan [9], several gene functions have been reported during the past 20 years. The first studies on PML-depleted mice revealed PML-dependent roles in apoptosis and myelogenic differentiation [9,10]. Subsequent studies, using knockout mice and cultured human cells have revealed important PML functions in a number of cellular processes, including senescence [11,12,13], angiogenesis [14,15,16], differentiation [17,18] and maintenance of genome stability [19,20,21,22] to name a few. Notably, several of these processes are part of a cell’s tumor suppression mechanisms. Accordingly, PML have been implicated in the pathogenesis of several forms of cancer, including breast, stomach, prostate, lung, colon and blood [23,24,25,26,27,28,29,30,31]. 

The PML bodies were originally described in 1984 by Bernstein et al. as nuclear dots that could be observed in HEp-2 cells following immunofluorescence labeling using serum from autoimmune patients [32]. Since their initial discovery these bodies have been referred to as PML nuclear bodies (PML NBs), nuclear domain 10 (ND10), PML oncogenic domains (PODs) and Kremer bodies [33,34]. In the present review we will primarily use the general term PML bodies which refer to all types of PML inclusions. Using conventional fluorescence microscopy PML bodies are typically detected as 4 to 30 nuclear dots, ranging from 0.1 to 1 µm in size. More advanced super-resolution or electron microscopy techniques have revealed that nuclear PML commonly forms a spherical shell with various PML body resident proteins localized either at their periphery or at their interior [35,36,37]. Although PML bodies appear relatively stable and immobile in most cells under normal physiological conditions, they are also known for their ability to change their morphology and protein composition in response to cell cycle progression, virus infection or various stress stimuli [38,39,40,41]. 

Cell division is vital for all living organisms, either for growth or for propagation. Cells from higher eukaryotes, such as mammals, undergo full disassembly of the nuclear envelope at entry into mitosis in order to facilitate transfer of a complete set of chromosomes to the newly formed daughter cells [42]. A consequence of such open mitosis is that nuclear and cytoplasmic components are mixed prior to each cell division, and this further leads to the requirement of proper sorting of cellular components to the cytoplasm or the progeny daughter nuclei at completion of mitosis, a process referred to as post-mitotic nuclear import. Unlike most nuclear compartments, PML bodies are not fragmented after entry into mitosis [43]. Rather, these bodies aggregate to form stable structures that persist during the entire mitosis. Consequently, PML bodies are deported from the nucleus to the cytoplasm each time a cell divides, and cytoplasmic disassembly during G1 is required for post-mitotic nuclear import of PML body components. Here we review the dynamics of PML bodies during mitosis and discuss implications of this behavior for post-mitotic nuclear import, cell fate decision and APL therapy.

## 2. PML Bodies and the Cell Cycle

### 2.1. PML Bodies in Interphase

PML bodies in interphase cells are predominantly detected within the nucleus but can also be detected in the cytoplasm of some cells, especially at early stages after mitosis. One characteristic feature of nuclear PML bodies is their capacity to recruit and sequester a diverse selection of nuclear proteins. Some of these proteins, such as DAXX, SP100, CBP and Small Ubiquitin-like Modifiers (SUMO), appear to be constitutively present in PML bodies. Others, for example p53, Rb, HP1, and a variety of stress-related proteins, are recruited in a cell type-dependent manner or in response to stimuli. A large proportion of PML-associated proteins is modified by SUMO conjugation and/or contains a SUMO interacting motif (SIM). Accordingly, several studies suggest the existence of a common PML body recruitment mechanism that depends on interactions between conjugated SUMO on one protein and a SIM on another [41,44,45,46,47]. Indeed, the PML protein itself contains both a SIM and at least three potential SUMOylation sites, and is thought to be the main protein responsible for protein recruitment to nuclear PML bodies. It has also been proposed that PML body self-assembly is mediated by PML through intermolecular SUMO-SIM interactions [46]. In agreement with this, tagged arrays of SUMO3 have been shown to phase separate into liquid droplets following mixing with tagged arrays of a SIM derived from PIASx [48]. However, several studies have also shown that seemingly morphologically normal PML bodies are generated from SUMOylation and SIM-defective PML proteins expressed in cells depleted of the wild type protein, suggesting that SUMOylated residues and the SIM domain are not essential for PML body assembly [49,50]. In addition, a recent study by Li et al. showed that also the variable C-terminal sequence of PML isoforms influence nuclear PML body formation and morphology [51].

Nuclear PML bodies generally distribute to the interchromatin space, and a number of reports show that their periphery frequently associate with specific chromatin elements or genomic loci [34,52,53,54,55,56,57]. These observations are in agreement with the fact that a large number of proteins that are recruited by PML bodies participate in chromatin metabolic processes, such as genome maintenance, DNA replication and gene expression [58,59,60]. In some extreme cases PML bodies appear to completely engulf certain chromatin elements. This is particularly evident in so-called alternative lengthening of telomere (ALT) cells, where recombination instead of telomerase is used for telomere lengthening. In these cells PML engages in formation of so-called alternative PML bodies (APBs) that contain telomere DNA in addition to a range of DNA repair proteins and telomere binding proteins [61,62,63]. In addition to chromatin, nuclear PML bodies have also been shown to target other nuclear structures, such as aggregated proteins [64,65,66,67,68], the nuclear membrane [69,70,71], lipid droplets [72], Cajal bodies [73,74] and nucleoli [75,76,77]. 

Several studies using live cell imaging of fluorescently tagged PML body-associated proteins have revealed a restricted local motility of these bodies under normal conditions, suggesting that once formed each body become restricted to a relatively small area of the nucleus [78,79,80]. It has not yet been clarified if the observed local motions are due to autonomous PML body motility or if it simply reflects motions of the local chromatin environment [81]. While PML bodies appear to be relatively stable in the nuclei of unperturbed healthy cells, they have the potential to undergo dramatic rearrangements under circumstances where cells are exposed to virus infection or various types of stressors [38,39,40,41,82]. These dramatic changes in PML body morphology are consistent with the emergent roles of PML, and many of the PML body resident proteins, in anti-virus and cell stress responses [83]. 

### 2.2. PML Bodies at the G2/M-Phase Transition

During interphase, the nuclear membrane provides a selective barrier that separates the nuclear interior from the cytoplasm. At entry into mitosis, the nuclear envelope breaks down leading to mixing of nuclear and cytoplasmic components. Several of the nuclear structures, including nucleoli, splicing factor compartments (speckles), lamina, and histone locus bodies disintegrate immediately following entry into mitosis [84,85,86,87,88]. This is also true for several cytoplasmic structures, for example endoplasmic reticulum and Golgi, which fragment into smaller membranous entities in order to facilitate proper distribution to daughter cells [89,90,91,92,93]. PML bodies, on the other hand, persist after entry into mitosis although their morphology and biochemical composition becomes dramatically altered (Figure 1). The first characterization of PML bodies in mitosis was published in 1999 by Everett et al. [34]. The authors of this paper showed that PML bodies in mitosis lack SUMO conjugates and certain PML body resident proteins such as SP100. They also observed increased phosphorylation of PML in mitotic cells compared to interphase cells. 15 years later Dellaire et al. investigated the dynamics of PML bodies in mitosis by expressing GFP-tagged PML-IV in U2OS cells. In this study, the mitotic PML bodies, which they designated mitotic accumulation of PML proteins (MAPPs), displayed increased motility and the ability to aggregate, leading to fewer but larger PML bodies [39]. In a subsequent study, Chen and co-workers carefully investigated PML body kinetics at entry into mitosis using U2OS cells stably expressing ECFP-PML-VI and EYFP-SP100 [94]. Based on thorough analysis of PML body motility relative to chromatin markers, they concluded that the increase in PML body directional movement at the G2/M transition is, in part, due to loss of chromatin tethering and is not merely a consequence of increased interchromatin space, or increased physical flow between the nucleus and the cytoplasm. Using mutational analysis of EYFP-tagged PML-I (the most abundant of the human PML isoforms) stably expressed in HaCaT cells, Lång et al. showed that mitotic and interphase PML bodies possess differences in their requirement for the zinc binding motifs within the PML TRIM [50]. In particular, point mutations that caused disruption of the second B-box domain of EYFP-PML-I led to complete abrogation of MAPPs and cytoplasmic assemblies of PML and nucleoporins (CyPNs), while the same mutations still supported formation of nuclear PML bodies in interphase cells.

The mixing of the nuclear and cytoplasmic environments during mitosis could potentially result in interactions between cytoplasmic and nuclear components. Accordingly, MAPPs have been observed to become specifically associated with early endosomes immediately following nuclear membrane breakdown [96]. In addition, a subset of MAPPs is observed to attach to mitotic spindle poles [96]. The functional significance of interactions between MAPPs and early endosomes, or between MAPPs and spindle poles, in mitosis, is currently not known. It is of interest to note, however, that a cytoplasmic PML isoform in mouse cells targets early endosomes to regulate TGFβ signaling in interphase cells [97], and that PML-I targets early endosomes in all phases of the cell cycle providing its nuclear localization signal (NLS) has been disrupted by mutagenesis [71]. 

### 2.3. PML Bodies at the M/G1-Phase Transition

PML bodies change interaction partners also as the cell progress from mitosis to G1. During this cell cycle transition they detach from early endosomes and spindle poles, and engage in interactions with components of the nuclear pore complexes (NPCs), forming cytoplasmic structures in G1 cells referred to as CyPNs [95,96,98] (Figure 1). They also associate with the microtubule network after exit from mitosis, and they exhibit microtubule-dependent movement as they gradually decrease in size and number due to disassembly and progressive transportation of their components to the nucleus [98]. 

Although PML bodies are too big to pass through the nuclear pore, a large body of evidence suggests that their presence in the cytoplasm is intermediate, and that their components are re-cycled back to the nucleus to support formation of progeny nuclear PML bodies after cell division. First, kinetic analysis show that cytoplasmic PML bodies decrease in size and number as the cell cycle progress after mitosis [39,98]. This is not due to proteolytic degradation of PML proteins, since cytoplasmic PML bodies disappear also in the presence of proteasome inhibitors [39]. Second, low, non-toxic concentrations of ATO are observed to stabilize cytoplasmic PML bodies, a phenomenon which is accompanied by reduced nuclear PML body regeneration in G1 cells [50]. Finally, cytoplasmic PML bodies are observed to recruit nuclear import components such as karyopherins and nucleoporins (NUPs), suggesting that proteins in these structures are targeted for nuclear import [95,98]. Notably, a systematic investigation of 20 different NPC components revealed that CyPNs have a preference for recruiting NUPs that face the pore channel and that are known to dynamically interact with nuclear import substrates that are passing through the NPCs. These so called peripheral NUPs failed to target CyPNs if their FG-repeat motif was disrupted or if importin β (a karyopherin nuclear import factor that facilitates PML nuclear import) was depleted [95]. Although it seems clear that PML bodies must be transported through the nuclear pores in pieces that are smaller than the CyPNs, the details by which these structures become disassembled into smaller units suitable for nuclear import remain poorly understood. 

It should be noted that conflicting results related to the role of MAPPs in regeneration of progeny PML bodies in G1 have been reported. Dellaire et al. reported the presence of chromatin-associated MAPPs that become entrapped inside newly formed nuclei after mitosis, an observation that led the investigators to propose that specific chromosome sites could be seeded by PML body fragments derived from the previous cell generation [39]. However, using both immunofluorescence labeling of endogenous PML in fixed cells and expression of fluorescently tagged PML in living cells, we have not been able to observe any trace of nuclear PML body fragments at early stages after chromatin encapsulation by the nuclear envelope [50,95,96,98,99]. This observation is consistent with the notion that nuclear PML bodies are formed de novo after mitosis without guidance by chromosome-associated PML body fragments inherited from the mother cell generation. The discrepancy could be due to a number of factors, including differences in cell lines used or differences in the expression level and behavior of fluorescently tagged PML isoforms.

The large size of PML bodies poses the advantage that one can visualize the spatio-temporal dynamics of PML body disassembly and nuclear import by the use of live cell tracking of cells expressing fluorescently tagged proteins. To our knowledge, PML is so far the only protein that has been observed, by use of direct visualization by fluorescence microscopy, to be targeted by nuclear import components in the cytoplasm. The observation that NUPs target cytoplasmic PML bodies after mitosis challenges the traditional view of the classical nuclear import pathway as a process that can be divided into two spatially separated steps; one step where the cargo protein binds to its nuclear import factor (importin β in the case of PML) at NPC distal sites and a second step where the import complex interacts with peripheral FG-containing NUPs within the NPCs [100]. The fact that many NUPs target PML within CyPNs immediately following nuclear import activation rather suggests a more active participation of NUPs in the formation of putative nuclear import complexes prior to interaction with and transport across the nuclear pores [95]. In future studies, it would be interesting to investigate a potential active role of NUPs in PML body disassembly and inhibition of PML aggregation after mitosis.

## 3. Implications of Mitotic PML Body Trafficking

### 3.1. Do PML Bodies Undergo a Liquid-to-Solid Transition at Entry into Mitosis?

A concept that has emerged in cell biology during the past 10 years is that several membrane-less organelles have liquid-like properties and form by phase separation. Such liquid-liquid phase separation is a physical process that occurs when certain compounds reach a super-saturated state, leading to spontaneous separation into a dense liquid phase that stably co-exists with the surrounding cytoplasm or nucleoplasm [101,102]. Nuclear PML bodies possess several characteristics that conform to a liquid-like state. They generally have a round appearance (which minimizes surface tension), they have the ability to undergo fusions and fissions [103], and they support rapid component exchange with the surrounding nucleoplasm [104]. Furthermore, the PML protein contains structural features that are consistent with a role of this protein in liquid-liquid phase separation. For example, PML contains a coiled-coil motif within its TRIM, which is a prominent protein-protein interaction motif commonly found in proteins that mediate liquid-liquid phase transition [102,105,106,107]. It also contains three SUMOylation sites that potentially provide multivalency, a feature that may facilitate phase separation through interactions with SIM-containing proteins [48]. 

While nuclear PML bodies detected during interphase can be characterized as liquids, the liquid-like properties seem to be lost during G2/M transition. This is evident from the reduced component exchange rate of MAPPs compared to nuclear PML bodies, and the ability to aggregate into amorphous multi-body structures without undergoing fusion as would be expected for bodies in a liquid state [39,94,95]. The transition from a liquid to a solid state during entry into mitosis could, in part, be due to extensive PML de-SUMOylation and the inability of these bodies to recruit SIM-containing proteins such as SP100 and DAXX. In agreement with this, interactions between multivalence scaffolds in the form of SUMO arrays and clients represented by SIM arrays have been shown to regulate liquid-liquid phase separation in a cell-free experimental system [48]. 

At exit from mitosis, the protein exchange mode of PML bodies seem to become reactivated, but in this phase the rate at which protein components leaves the bodies is much higher compared to the rate at which proteins are entering the bodies. Consequently, PML bodies in the form of CyPNs gradually disassemble following exit from mitosis. It is possible that the nuclear import components participate in re-solubilizing the cytoplasmic PML bodies at this stage of the cell cycle. In this respect, it is of interest to note that FG-repeat-containing NUPs, which are observed to encapsulate cytoplasmic PML aggregates at exit from mitosis, have the ability to phase separate into hydrogels, a property that facilitates solubilization and transport of nuclear import complexes [108]. The PML protein may therefore occupy different physical phases depending on the cell cycle stage: a liquid phase during interphase, a solid phase during mitosis and a hydrogel phase during nuclear import (Figure 2). 

### 3.2. The Role of Mitotic PML Bodies in Asymmetric Cell Division and Cell Fate 

During embryonic development, tissue regeneration or homeostasis, stem cells and progenitor cells divide and proliferate in a highly regulated manner in order to balance stem cell renewal and tissue formation. Several studies have implicated the PML protein in these fundamental cellular processes. For example, disruption of PML through trans-dominant PML-RARα activities may, in part, account for the defects in myelogenic differentiation that lead to APL. In agreement with this, PML has been shown to play a critical role in the maintenance of hematopoietic stem cells [26,109]. Furthermore, PML has been reported to regulate differentiation and/or regeneration in several non-hematopoietic tissues, including brain [18,110], mammary gland [111], bone [112] and muscle [113]. Lastly, PML has been shown to regulate the balance of self-renewal and differentiation in pluripotent stem cells [114]. 

Mechanistically, PML seems to have a large potential of regulating cell fate and differentiation at the level of cell division. Strong evidence for this came from a study by Ito et al. showing that a PML–PPAR-δ-regulated mechanism controls maintenance of hematopoietic stem cells through activation of fatty acid oxidation [17]. By using Tie2 and CD48 as markers for stem cells and linage committed cells, respectively, they demonstrated decreased asymmetric cell division concomitant with stem cell exhaustion following depletion or inhibition of PML and/or PPAR-δ. Since PML bodies are inherited from the mother to the daughter cells during cell division, these bodies may become asymmetrically apportioned into the cytoplasm of newly formed G1 cells. Indeed, asymmetric segregation of PML bodies was recently demonstrated in two-dimensional sheets of collectively migrating keratinocytes [99]. In this experimental system, cell divisions were found to be highly polarized and frequently occurred along the same axis as the direction of collective cell migration. Careful examination of these cell divisions revealed that the nucleus repositioned itself towards the migrating front prior to cell division. Consequently, the PML bodies exhibited a higher tendency for segregating towards the daughter cell derived from the front. The cytoplasmic lysosomes, on the other hand, which was observed to occupy the opposite side of the nucleus prior to cell division, showed a preference for the daughter cell derived from the rear part of the mother cell. Notably, the ability of PML bodies to become asymmetrically partitioned during mitosis increased significantly in stem cell-enriched primary human keratinocytes, and the cells receiving the majority of PML bodies generally exhibited increased stemness [99]. Thus, PML bodies can be added to the list of cellular components that regulate stemness and that have the potential of becoming asymmetrically partitioned during stem cell division. 

### 3.3. The Role of Mitotic PML Bodies in Acute Promyelocytic Leukemia (APL) Therapy 

The reciprocal t(15;17)(q22;q21) chromosomal translocation and its link to APL was first described in the early nineties [1,2,3]. The PML-RARα fusion oncoprotein, which is the product of this chromosomal aberrancy, function as a transcriptional repressor in APL cells, leading to aberrant expression of genes involved in blood cell differentiation [115,116,117]. Indeed, one of the main characteristics of APL is an accumulation of immature promyelocytes in the bone marrow and peripheral blood. The discovery of PML-RARα as the main cause of APL did not only lead to important insight into the cause of the disease. It also explained, at least in part, why therapies involving the RARα ligand all-trans retinoic acid (ATRA) resulted in a dramatic improvement of APL cure rates during the late eighties [118,119,120,121]. Shortly after discovery of the ATRA-based therapy and the PML-RARα fusion protein, a second drug, namely arsenic trioxide (ATO), was also shown to cure APL, even as a single agent [122,123,124]. Remarkably, while ATRA targets the RARα moiety of the oncoprotein, ATO seems to abrogate PML-RARα functions through interactions with zinc binding motifs present within the PML TRIM [125,126].

APL pathogenesis and cure is also reflected by changes in PML body morphology. Expression of PML-RARα transforms PML bodies into a microspeckled pattern, a phenotype that is reversed by ATRA-mediated degradation of PML-RARα [127,128]. Treatment of APL cells with ATO also leads to PML-RARα degradation, but the presence of this drug does not fully restore normal PML body integrity [129,130,131]. One possible reason why ATO fails to restore PML body morphology could be that this drug has the ability to target both PML-RARα, as well as PML expressed from the non-rearranged allele [132]. At the molecular level ATRA and ATO are thought to cure APL mainly through activation of proteasome and/or autophagy-dependent degradation of PML-RARα [129,130,131,133]. In a recent study, Lång and co-workers showed that ATO also have the potential to prevent nuclear PML and PML-RARα functions by restricting their nuclear recycling after mitosis [50]. In this study, low pharmaceutically relevant concentrations of ATO was found to significantly delay the transition of PML and PML-RARα from CyPNs to nuclear PML bodies after mitosis, leading to accumulation of these proteins in the cytoplasm. As a consequence, ATO-based therapy may lead to delayed recruitment of PML and PML-RARα to nuclear transcription sites after cell division. Notably, two mutations in the second B-box domain of PML, which previously had been identified in ATO-resistant APL patients, led to failure of PML to accumulate in the cytoplasm after mitosis due to the inability of these mutants to support formation of MAPPs and CyPNs during entry into mitosis [50,134]. Thus, the ability of PML bodies to persist as MAPPs and CyPNs after entry into mitosis may influence ATO-based APL therapy.

### 3.4. Cell Compartment Sorting by the Nuclear Localization Signal during Post-Mitotic Nuclear Import

Although PML is mainly detected in the nucleus, it has also been reported to have functions in the cytoplasm. In a study published by Lin et al., it was shown that PML targets early endosomes in order to regulate TGFβ-mediated signaling [97]. Dysregulation of this PML-TGFβ signaling pathway has been reported to be involved in the development of invasive prostate cancer [23]. Cytoplasmic PML has also been shown to regulate autophagy-dependent apoptosis and calcium transport through interactions with mitochondria and the endoplasmic reticulum [135,136]. Since PML proteins that are sequestered within CyPNs account for the majority of PML detected by fluorescence microscopy in a growing cell culture, it is tempting to speculate that this population of visible PML proteins could participate in the reported PML-dependent cytoplasmic functions. However, PML proteins that have been relocated to the cytoplasm through mitosis do not exhibit a significant association with endosomes, mitochondria or endoplasmic reticulum and little evidence have so far been presented which suggest that PML proteins within CyPNs become engaged in cytoplasmic functions prior to their nuclear import. Rather, cytoplasmic PML functions are more likely to be carried out by the alternatively spliced PML isoforms that lack the central NLS. In agreement with this, cytoplasmic defects in TGFβ signaling and calcium transport, which was observed in PML-depleted mouse cells, were effectively rescued by cytoplasmic PML splice variants [97,135]. A caveat of this latter theory, however, is that all PML isoforms discovered to date (including the cytoplasmic once) harbor a functional TRIM and therefore have the potential to form strong protein-protein interactions with all of the other PML isoforms. Thus, it is not clear how the low levels of cytoplasmic PML isoforms avoid being relocated to nuclear PML bodies through heterodimerization with the far more abundant nuclear isoforms [70,137]. It is possible that hitherto undiscovered cellular mechanisms facilitate dedication of proteins to a cytoplasmic or nuclear state, immediately following protein synthesis. 

It is worth noticing that nuclear PML isoforms (with the exception of PML-II) target endosomes in the cytoplasm following distortion of their central NLS by point mutations [71]. In addition, the endogenous pool of PML proteins target the endosomes during mitosis, the period of the cell cycle where components of the nuclear import machinery have been inactivated or repurposed to perform mitotic functions [96]. It is conceivable, based on these observations, that the NLS in conjunction with nuclear import factors functions as a molecular switch that in its “on-state” directs PML function to the nucleus and in its “off-state” directs PML to the endosome vesicles in the cytoplasm (Figure 3). Similar strategies for masking or deletion of the NLS for the purpose of redirecting a protein to other compartments are well known for several proteins [138,139,140,141,142].

## 4. Conclusions and Outlook

PML bodies seem to behave differently from most nuclear and cytoplasmic compartments in that they do not undergo extensive disassembly during mitosis. In fact, instead of fragmenting into smaller units, which potentially could facilitate distribution of subcomponents among daughter cells, these bodies choose to do the opposite by aggregating into larger and fewer bodies. One possible reason for this behavior may be that PML bodies function as transport vehicles during mitosis. It is clear from several studies cited in the present review that PML bodies in the form of MAPPs and CyPNs influence the transport of PML proteins from the nucleus to the cytoplasm during mitosis. Indeed, PML proteins in primary human epidermal stem cells typically aggregate into only two or three large bodies that frequently segregates asymmetrically to only one of the two daughter cells [99]. One remaining question, though, is whether PML bodies have the ability to co-aggregate with and thus influence the fate of other proteins during mitosis. It would be interesting, for example, to know if PML bodies have the ability to selectively influence nucleo-cytoplasmic translocation of nuclear aggregates consisting of polyglutamine (polyQ)-expanded proteins. Such protein inclusions, which are involved in the pathogenesis of several neurological disorders, are known to co-localize with PML bodies, and PML has been shown to promote their degradation [64,65,68,143,144,145]. Interestingly, PML also forms similar inclusions that contain viral capsid proteins following infection with varicella-zoster virus [146]. Future studies should investigate the fate of such protein aggregates during cell division in the presence and absence of PML. 

The behavior of PML bodies during mitosis also provides unique opportunities to investigate molecular mechanisms related to reactivation of nuclear import following mitosis. It is well known that components of the nuclear import machinery disassemble and become inactivated, or repurposed for mitotic functions, during cell division. Thanks to the significant size of PML protein inclusions at transition from mitosis to G1, the assembly dynamics of PML nuclear import complexes can be readily visualized by the use of fluorescently tagged proteins in living cells. Using PML as a model protein it may be possible to investigate many of the important unanswered questions related to post-mitotic nuclear import. 

## Figures and Tables

**Figure 1 cells-08-00893-f001:**
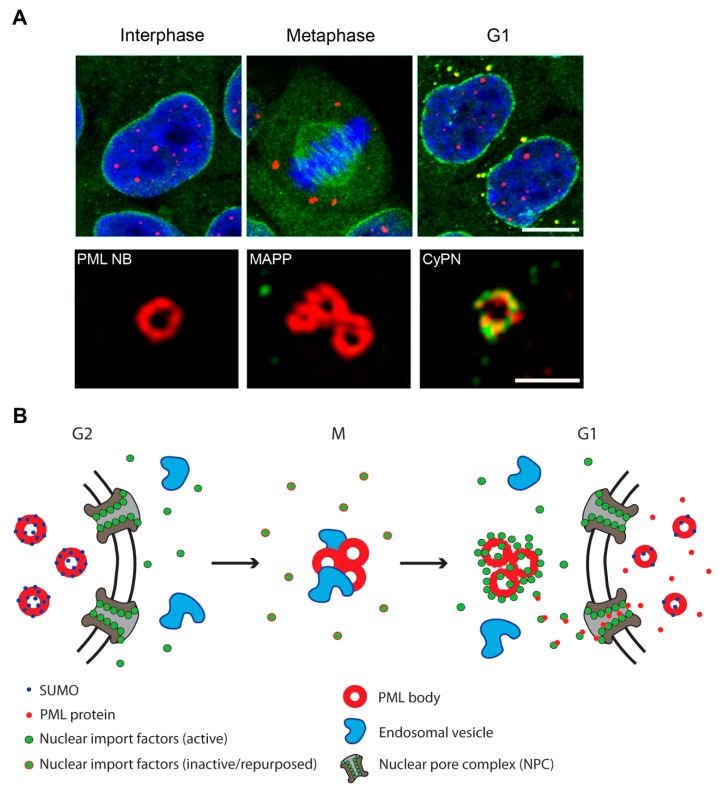
PML bodies at different stages of the cell cycle. (**A**) Upper panel: Confocal images of fixed HaCaT cells. DAPI (blue), PML (red), Importin β (green). Scale bar 10 µm. Lower panel: Super-resolution structured illumination microscopy (SIM) images showing PML bodies in fixed HaCaT cells. PML (red), FG-containing NUPs (green). Scale bar 1 µm. PML NB (PML nuclear body), MAPP (mitotic accumulation of PML proteins), CyPN (cytoplasmic assemblies of PML and nucleoporins). Images have been reproduced from Lång et al. ([95], www.tandfonline.com) (**B**) Schematic illustration of PML bodies in mitosis. At entry into mitosis, PML bodies aggregate, become de-SUMOylated, and associate with endosomal vesicles. Nuclear import factors become inactivated or repurposed during mitosis and re-activated at exit from mitosis to engage in import of nuclear components (please see main text for more details).

**Figure 2 cells-08-00893-f002:**
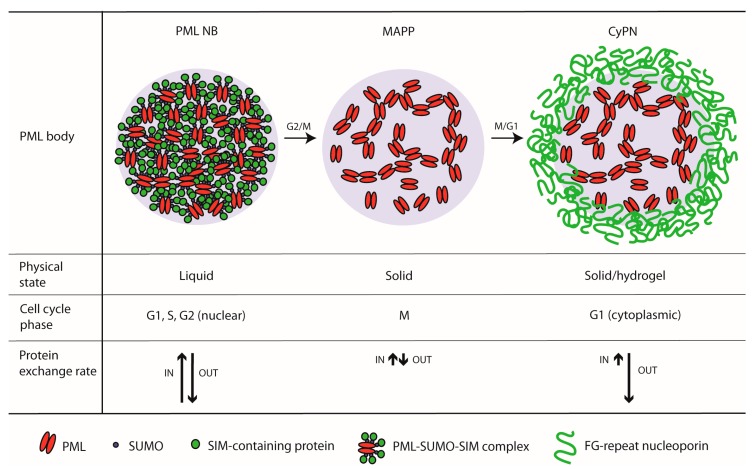
PML body biophysics at different stages of the cell cycle.

**Figure 3 cells-08-00893-f003:**
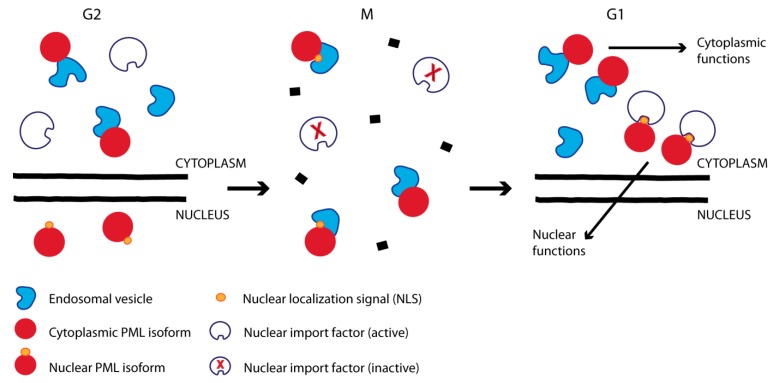
Cell compartment sorting of cytoplasmic and nuclear PML isoforms during progression through mitosis: NLS-containing PML isoforms are directed to the nucleus during interphase but associate with endosomes in mitosis due to nuclear import inactivation. Cytoplasmic PML splice variants, which lack an NLS, retain their association to cytoplasmic compartments (such as endosomes) throughout the cell cycle.

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
