# Peer review of "PML Bodies in Mitosis"

_cells, 2019, doi:10.3390/cells8080893_

Round 1

Reviewer 1 Report

Promyelocytic Leukemia Bodies in Mitosis

 by Lång et al.

In their review, Anna Lång and collagues summarize recent advances in our understanding of PML nuclear body dis/assembly and maintenance, particularly during cell division. The group has previously pioneered potentially new mechanism(s) of nuclear import by careful spatio-temporal analysis of PML nuclear bodies and nuclear import factors behavior at all stages of mitosis and early G1 cell cycle phase. In the current review all these new observations are discussed with respect to a newly proposed molecular mechanism of nuclear import, asymmetric stem cell division/decision and cell fate. This is a timely summary of their findings with two new molecular models of cell compartment and NPC sorting. The review is nicely written and the figures nicely illustrate the authors’ new views. The messages will be of broad interest to the broad readership of nuclear organization and the new model(s) will shape future developments in the field of nuclear body genesis, nuclear trafficking and PML’s role in stem cell fate.

A few minor issues to further improve the manuscript are suggested by this reviewer:

(1) line 14:

The broad term “cancer therapy” appears to big in the abstract as only APL is discussed in the main text. Are there other types of cancer related to ther new observations discussed which might be mentioned in the main text? Otherwise “APL” would be more appropriate.

(2) line 24:

“several which” should read “several of which”

(3) line 45:

When the shell structure of PML bodies are discussed, this paper must be cited as well:

Lang et al. J Cell Sci. 2010 Feb 1;123(Pt 3):392-400. doi: 10.1242/jcs.053496.

(4) line 50:

Not all readers may be familiar with the term “open mitosis”. Authors may want to briefly explain it (i.e. ”where the nuclear envelope breaks down before the chromosomes separate”)

(5) line 68:

Reference 37 (Weger et al.) may be inappropriate in this context as a GFP fusion of Topors is not recruited to endogenous PML bodies as reported in this paper. Authors could cite Brown et al.which shows that the SIM in PIAS1 is required to targed this protein to PML bodies.

Brown JR, Conn KL, Wasson P, Charman M, Tong L, Grant K, McFarlane S,

Boutell C. 2016. SUMO ligase protein inhibitor of activated STAT1 (PIAS1) is a

constituent promyelocytic leukemia nuclear body protein that contributes to the

intrinsic antiviral immune response to herpes simplex virus 1. J Virol

90:5939 –5952. doi:10.1128/JVI.00426-16

(6) line 73:

Banani et al. used the SIM motif derived from PIASx, not PML’s SIM. Please correct.

(7) Along these lines of liquid-liquid phase sepearation (LLPS):

Although there is no direct evidence for phase separation at PML bodies in live cells, authors may want to discuss briefly what this newly recognized biophysical mechanism could mean in the context of MAPPs and CyPNs, as they are nicely depicted in Fig. 1A. PML NBs in interphase are likely to form compartments based on LLPS as their component parts more or less rapidly exchange with the nucleoplasm (Weidtkamp-Peters et al., doi: 10.1242/jcs.031922). However, MAPPs do not exhibit PML exchange with the nucleo-cytoplasm during mitosis (Dellaire et al., J Cell Sci. 2006 Mar 15;119(Pt 6):1034-42.) and hence may “only” act as solid scaffolds (for nuclear import factors?) without LLPS. Is it possible that MAPPs become more fluid (LLPS) again during their transition into CyPNs? Authors may want to think along these lines and suggest the role of LLPS. For example, FG repeat containing NPC factors provide a large number of polyvalent modules fpr LLPS an CyPNs. As LLPS has become a major issue in cell biology this review should not go without putting this new concept into relationship with the auhtor’s exciting new findings.

(8) Chen et al. must be cited and discussed

One important paper on PML body behavior during mitosis is missing and must be included: Chen YC1, Kappel C, Beaudouin J, Eils R, Spector DL. Live cell dynamics of promyelocytic leukemia nuclear bodies upon entry into and exit from mitosis. Mol Biol Cell. 2008 Jul;19(7):3147-62. doi: 10.1091/mbc.E08-01-0035. Epub 2008 May 14.

Authors of this paper observed that MAPPs become trapped within newly formed nuclei. This is somewhat inconsistent with the model proposed in Fig. 1B. The discrepancy may be related to the use of overexpressed FP-tagged PML. Anyway, this issue should be discussed in the review.

Reviewer 2 Report

 The authord describe PML bodies functions, localizations and dynamics across mitotis and introduce their potential role in cancer therapy.

The first part is organized and provides a clear review of the topic, while the second one is not complete and does not discuss the interplay between nuclear bodies and tumor hallmarks (e.g. BRCA genes). Hypothesis and speculations on PML bodies in cancer therapies are missing and should be added.

The reference list is not up to date, with the overwhelming majority of cited articles published at least 5 years ago and back before 2000. The authors do not discuss recent potentially important developments in this field, for example data linking POT1 and ALT-assoiated PML bodies (Episkopou et al. Mol Cell 2019) and others.

Round 2

Reviewer 2 Report

it is a long review article well written and intensive presented. 

Some redundancies are present. I have read the article with great interest and pleasure.

The paper is acceptable for pubblication.